# Novel NIR-Phosphorescent Ir(III) Complexes: Synthesis, Characterization and Their Exploration as Lifetime-Based O_2_ Sensors in Living Cells

**DOI:** 10.3390/molecules27103156

**Published:** 2022-05-14

**Authors:** Ilya S. Kritchenkov, Vitaliya G. Mikhnevich, Victoria S. Stashchak, Anastasia I. Solomatina, Daria O. Kozina, Victor V. Sokolov, Sergey P. Tunik

**Affiliations:** Institute of Chemistry, Saint-Petersburg State University, Universitetskii pr., 26, 198504 St. Petersburg, Russia; i.s.kritchenkov@spbu.ru (I.S.K.); st076006@student.spbu.ru (V.G.M.); st076141@student.spbu.ru (V.S.S.); nastisol@gmail.com (A.I.S.); st055671@student.spbu.ru (D.O.K.); v.sokolov@spbu.ru (V.V.S.)

**Keywords:** phosphorescence, iridium complexes, oxygen sensors, bioimaging

## Abstract

A series of [Ir(**N^C**)_2_(**N^N**)]^+^ NIR-emitting orthometalated complexes (**1**–**7**) has been prepared and structurally characterized using elemental analysis, mass-spectrometry, and NMR spectroscopy. The complexes display intense phosphorescence with vibrationally structured emission bands exhibiting the maxima in the range 713–722 nm. The DFT and TD DFT calculations showed that the photophysical characteristics of these complexes are largely determined by the properties of the metalating **N^C** ligands, with their major contribution into formation of the lowest S_1_ and T_1_ excited states responsible for low energy absorption and emission, respectively. Emission lifetimes of **1–****7** in degassed methanol solution vary from 1.76 to 5.39 µs and show strong quenching with molecular oxygen to provide an order of magnitude lifetime reduction in aerated solution. The photophysics of two complexes (**1** and **7**) were studied in model physiological media containing fetal bovine serum (FBS) and Dulbecco’s Modified Eagle Medium (DMEM) to give linear Stern-Volmer calibrations with substantially lower oxygen-quenching constants compared to those obtained in methanol solution. These observations were interpreted in terms of the sensors’ interaction with albumin, which is an abundant component of FBS and cell media. The studied complexes displayed acceptable cytotoxicity and preferential localization, either in mitochondria (**1**) or in lysosomes (**7**) of the CHO-K1 cell line. The results of the phosphorescence lifetime imaging (PLIM) experiments demonstrated considerable variations of the sensors’ lifetimes under normoxia and hypoxia conditions and indicated their applicability for semi-quantitative measurements of oxygen concentration in living cells. The complexes’ emission in the NIR domain and the excitation spectrum, extending down to ca. 600 nm, also showed that they are promising for use in in vivo studies.

## 1. Introduction

Molecular oxygen is one of the most important components of aerobic biological objects, without which their vital activity is impossible. Deviation of its content from the norm can cause various pathologies, or indicate their development [1,2,3]. Thus, determination of the O_2_ concentration in biological systems is extremely important for monitoring their physiological status. In this respect, the application of luminescent molecular and nanosized sensors based on phosphorescent emitters has an undoubted advantage, because in the presence of molecular oxygen their emission is effectively quenched, providing clearly detectable variations in emission intensity along with the quantitative dependence of an excited state lifetime on oxygen concentration. It should be noted that, in recent years, the advanced technique of phosphorescent lifetime imaging (PLIM) has found a wide range of applications as an important instrument for monitoring the oxygen concentration in biological systems [4,5,6,7,8,9,10]. Unlike the oxygen sensing methods based on quantitative considerations, PLIM measurements do not need external or internal standards and the sensor response depends neither on the local sensor concentration nor on the optical properties of the sample under study that exclude the distortion of the results by these biasing factors [5,7,11,12,13].

Among the available phosphorescent chromophores, the complexes of transition metals such as Ru(II), [10,14,15,16,17] Pd(II), [10,17] Re(I), [10,17] Ir(III), [10,14,15,16,17], and Pt(II) [10,14,16,17] are most often used for monitoring oxygen concentration, since many of these coordination compounds exhibit bright luminescence in the near infrared (NIR) region and show substantial response of their emission intensity and lifetime values to the presence of molecular oxygen. In this study we paid particular attention to obtaining iridium emitters, which exhibit phosphorescence in the so-called transparency window of biological tissues, i.e., in the NIR range that can expand the possibilities of their effective application for in vivo bioimaging [10]. Luminescent iridium complexes are of particular interest, since the availability of six coordination positions at Ir(III) ion makes possible a rather wide variation in the nature of coordinated ligands, allowing fine tuning of the complexes’ photophysical characteristics, as well as the introduction of required functional groups into the ligand’s periphery for subsequent emitters’ functionalization (vectorization, binding to biomolecules, imparting solubility in water, and biocompatibility, etc.). 

As a rule, many iridium complexes of the [Ir(**N^C**)_2_(**N^N**)]^+^ type are characterized by high chemical inertness and stability with respect to photobleaching, high luminescence quantum yields, and relatively low toxicity [18,19,20,21,22,23,24,25]. Therefore, we used a number of new cyclometallating **N^C** and chelating diimine **N^N** ligands in the synthesis of the target Ir(III) compounds. Variations in the composition of the ligands’ aromatic systems and in the donor/acceptor ability of substituents allowed for obtaining effective NIR phosphors with an appreciable lifetime response to the changes in oxygen concentration in solution, which were then used in PLIM bioimaging experiments. In order to test the applicability of the target compounds for oxygen sensing in biological systems, it was important to evaluate the effects of the other external stimuli (temperature, salinity, and interaction with biomolecules) on the emitters’ photophysical parameters. The corresponding experiments were carried out in model physiological media containing Dulbecco’s Modified Eagle Medium (DMEM) and fetal bovine serum (FBS), because these mixtures are commonly used as growth media for cell monolayers and 3D spheroids, as well as for cell cultures’ incubation with labels and sensors of various natures. The interaction of obtained iridium complexes with a major component of physiological media (albumin) was also studied. It was shown that the interaction of the hydrophobic molecules of iridium complexes with albumin determines the sensors’ sensitivity to oxygen and should always be taken into account while preparing the calibration curves for quantitative oxygen measurements in biological systems. For a couple of the obtained complexes, the corresponding calibrations were obtained and the PLIM experiments with the CHO-K1 cells demonstrated the sensors’ applicability for monitoring oxygen concentration in living cells.

## 2. Results and Discussion

Two groups of [Ir(**N^C**)_2_(**N^N**)]^+^ NIR-emitting orthometalated complexes were prepared using the standard synthetic procedure [26,27,28,29,30,31,32] shown in Figure 1. In the first group of complexes (**1–4**), the orthometalating **N^C** ligand remained unchanged, whereas the nature of the diimine **N^N** ligand was varied considerably to investigate the effect of its aromatic system on the emission characteristics of the final products. In contrast, the complexes of the second group (**5–7**) contained the same diimine ligand and the electronic characteristics of the **N^C** aromatic system were modified by the introduction of electron-withdrawing (-COOH) and electron-donating (-OMe) substituents to the **N**- and **C**-coordinated fragments. These modifications were aimed at variations in the complexes’ emission properties, as well as at tuning the complexes’ solubility in aqueous solutions for application in biological experiments.

These complexes were thoroughly characterized by using elemental analysis, HR ESI mass-spectrometry, and ^1^H NMR spectroscopy (1D and 2D COSY and NOESY). However, we were not able to obtain single crystals suitable for X-ray crystallographic analysis. The obtained analytical and spectroscopic data (see Appendix A, where the prefix S denotes tables and figures presented in the electronic supporting information file) were in complete agreement with the coordination environment typical for octahedral iridium complexes of this type [26,27,28,29,30,31,32,33]. In these structural patterns, two coordinated **N^C** ligands give *trans-* and *cis-*dispositions of their **N** and **C** functions, respectively, whereas the chelate diimine ligand occupies the two sites at the coordination octahedron, which are *trans* to the **C** functions of the metalating ligands. The DFT optimization of the complexes’ ground state structure, as shown in Figure 1 and Appendix A, confirms the structural assignment made on the basis of the spectroscopic data.

All complexes were luminescent in methanol solutions; their photophysical characteristics are summarized in Table 1. Absorption, excitation, and emission spectra are shown in Appendix A and Figure 2, respectively. The maxima of the low energy absorption bands of these compounds were in the range of 520–540 nm, as shown in Appendix A, with the tails extending down to 650 nm. According to the results of the DFT calculations (Appendix A, Appendix A), these bands may be assigned to the transitions, with the major contribution of the ligand (**N^C**)-centered (LC) character mixed with the interligand (ILCT) and metal-to-ligand (MLCT) charge transfer. Note that the diimine ligand orbitals did not take part in the formation of these low energy singlet excited states. The complexes display essentially similar emission band profiles in the NIR region, with the band maxima in the range of 713–722 nm, as shown in Figure 2. The bands showed vibrational structure with the spacing of ca. 1250 cm^−1^ that fit well the magnitudes of vibration frequencies typical for the ligands’ aromatic systems. This observation also pointed to the domination of the ligand-centered character in the nature of the emissive excited state. The results of the DFT and TD DFT calculations for the key representatives of these emitters (complexes **1** and **7**) (Appendix A, Appendix A) confirmed this conclusion and provided the values of emission band maxima, which were in a very good agreement with the experimental data. The emission lifetimes for all complexes fell in the microsecond domain and showed approximately an order of magnitude reduction in aerated solutions compared to the degassed ones (Table 1). This indicated that emission occurs from the triplet excited state, i.e., phosphorescence. 

The obtained experimental data and the results of the theoretical calculations led to several important conclusions: the **N^N** ligands do not participate in emissive excited state formation and negligibly affect the magnitude of the T_1_-S_0_ energy gap, and the properties of the **N^C** benzothienyl-phenanthridine aromatic system have a key impact on the emissive energy gap. It is also worth noting that essentially different substituents (cf. -COOH and -OMe) inserted into the benzothienyl-phenanthridine fragment of the **N^C** ligands did not perturb the emission energy, evidently due to a large “capacity” of this electronic reservoir. The phosphorescence quantum yields were rather high for the NIR-emitting phosphors (10.3–20.5% in degassed solution) and fit well with the other data obtained earlier for complexes of this type [10,26,27,28,30,31,32,33,34,35]. Interestingly, the emission efficiency (QY) was nearly the same for complexes **1–6** and sharply different for **7**, increasing for ca. 60% compared to the other complexes. The data on radiative and nonradiative rate constants (Table 1) indicate that in the case of **7** one can observe a substantial decrease in the rate of nonradiative emissive excited state relaxation, while radiative rate constants are very similar for the emitters **1** and **5–7****,** which contain the same **N^N1** ligand and differ only in the structure of the **N^C** ligands. This observation may be explained by the “rigidification” of the molecular structure in **7** caused by the introduction of four -OMe substituents in the **N^C4** ligands, which evidently limits possible distortions in the emissive triplet compared to the ground state and thus reduces the so-called Huang-Rhys factor [36,37], which determines k_nr_ magnitude of phosphorescent emitters. Note that the intramolecular noncovalent interactions of OMe substituents with the other components of ligand environment were also observed in the NOESY spectra of **7** (see Appendix A).

The obtained complexes displayed a rather high quantum yield in the degassed methanol solution and substantial sensitivity of emission parameters (QY and τ, as shown in Table 1) to the presence of oxygen that makes them promising candidates for application as oxygen sensors in biological systems. The complexes **1** and **7** with the highest emission intensity were chosen to explore their applicability in the studies of cells’ oxygenation. To increase the compounds’ solubility in physiological media, we prepared starting aqueous solutions of **1** and **7** containing polyethylene glycol (PEG 200), which were then used for cells’ incubation together with the standard components of cell culture media, fetal bovine serum (FBS), and Dulbecco’s Modified Eagle Medium (DMEM).

From the viewpoint of photophysical measurements, these media also simulated well the sensor microenvironment in cell cultures [27,28,38,39,40], and we carried out Stern-Volmer calibration for **1** and **7** using DMEM-FBS-PEG(200) solutions, as shown in Figure 3 and Table 2. Quite expectedly, because of the essentially similar nature of the chromophore, both complexes showed nearly equal magnitudes of K_SV_, which, however, were very different from those observed in methanol solution (cf. the lifetime difference for degassed and aerated solutions in aqueous media and methanol, Table 1 and Table 2). The reason for such different behavior was the interaction of rather hydrophobic sensor molecules with the components of DMEM/FBS mixtures, of which the bovine serum albumin (BSA) molecules were the most probable candidates for the sensors’ absorption in their hydrophobic pockets, thereby changing the photophysics of the iridium emitters [27,28,41,42,43,44,45,46]. To verify this hypothesis, we performed lifetime measurements for **1** and **7** in aerated and degassed aqueous solutions containing BSA as the only biological molecule (see Table 2). Under the limit of experimental uncertainty, the values obtained matched those found in the complex {DMEM/FBS} mixture, a clear indication of the emitters’ interaction with BSA to give noncovalent {BSA/complex} conjugates.

We also studied the composition of the adducts formed in this reaction by using gel-permission chromatography (GPC). Typical GPC traces of the {BSA+**1**} and {BSA+**7**} reaction mixtures are shown in Figure 4. The monitoring of the products’ absorption at 215 nm (Figure 4A), where the BSA extinction coefficient absolutely dominated over those of **1** and **7**, indicated that the formation of the {BSA/**1**} and {BSA/**7**} adducts did not provoke BSA aggregation, as could be expected for the adducts of rather large hydrophobic molecules [42].

The GPC traces recorded at the wavelength of 380 nm, where the extinction coefficients of the complexes were substantially larger than that of BSA, indicated that the mixture still contained free BSA, did not show the peaks of free complexes, and displayed the peaks corresponding to the components, with the molecular mass slightly higher than that of free BSA. The latter signals may be reasonably assigned to the {BSA/Complex} adducts. The photophysical and chromatographic data provided above testified in favor of the complete embedding of the complexes into albumin pockets in solutions commonly used for cell culturing. These adducts were stable in the model physiological media, displaying reliable photophysical parameters, and thus could be used for oxygen-sensing in biological samples. For application in this type of study, we evaluated the cytotoxicity of these complexes on living Chinese hamster ovary cells (CHO-K1), using MTT assay. As in the photophysical study, the experiments on the hydrophobic complexes **1** and **7** were solubilized in a DMEM-FBS-PEG(200) mixture. Thus, both the contents of PEG-200 in the cell culture media and the concentration of complexes impacted the overall toxicity of the probes. To determine the optimal complex—PEG-200 ratio and concentrations for the bioimaging experiments and maximize cell viability—we determined the toxicity of the complexes in the presence of a constant concentration of PEG-200 (Figure 5A), and the toxicity of PEG-200 at different concentrations in the absence and presence of a constant concentration of complexes (Figure 5B). Upon 24 h of incubation, PEG-200 demonstrated moderate toxicity (cell viability ca. 80%) at concentrations up to 2.5%. Compared to the PEG-200 toxicity, complex **7** displayed a minor effect on the viability and proliferation of CHO-K1 cells within the tested concentration range (1–25 µM). Complex **1**, however, revealed pronounced cytotoxicity, starting from 5 µM (cell viability less than 50%), and even increased the toxicity of PEG-200. For the further microscopy experiments, we used the following concentrations of complexes in the growing media: 5 µM of **1** with 1.5% of PEG-200 and 10 µM of **7** with 2.5% of PEG-200.

These concentrations allowed the obtaining of images of subcellular sensors’ distributions in living CHO-K1 cells (Figure 6) and the carrying out of PLIM measurements during an acceptable time interval (Figure 7). The co-localization studies (Figure 6) with the corresponding organelle-specific fluorescent dyes—Hoechst 33324 (nuclei), BioTracker 405 Blue Mitochondria Dye (mitochondria) and LysoTracker Green DND-26 (lysosomes)—showed slightly different intracellular distributions of **1** and **7**. The calculated Pearson’s (P) and Mander’s (M1) overlap coefficients for the complexes and the mito- and lyso-specific dyes suggested the preferential mitochondria localization of **1** and the dominating localization of **7** in lysosomes. This difference can be assigned to the structural features of the ligand environment in these complexes. Hydrophobic ligands and the positive charge of **1** increased its affinity to mitochondria, whereas the presence of hydrophilic (-COOH) substituents in the **N^C4** ligands of **7**, together with the possibility of deprotonation of the carboxylic group and the loss of positive charge of the complex on the whole, made mitochondria staining for **7** less probable and promoted endosomal localization. The preferable distribution of complex **1** in mitochondria could also explain its high cytotoxicity, *vide supra*. Both complexes demonstrated negligible co-staining with nuclei dye, indicating no penetration into the cell nucleus.

PLIM images obtained after incubation of the CHO-K1 cells with **1** and **7** under normoxia and hypoxia are shown in Figure 7. The complex **1** localized mainly in mitochondria displayed clearly discernible lifetime distributions in aerated and deaerated cells, the average lifetime values being very close to those obtained in cuvette using model physiological media (DMEM/FBS/PEG-200) (Table 2). The PLIM experiment with **7** also showed considerably different lifetime distributions for the cells under normoxia and hypoxia, but the average values displayed larger deviations (of ca. 0.5 µs) from the values obtained in cuvette, which may be explained by the different microenvironments of the sensor molecules in lysosomes and in the model media. Nevertheless, even this complex may be used for the qualitative estimation of oxygenation in biological samples to distinguish normoxic and hypoxic areas.

## 3. Conclusions

We prepared seven novel cationic iridium [Ir(**N^C**)_2_(**N^N**)]^+^ complexes, which displayed rather strong NIR emission (λ_em_ > 710 nm, QY from 10.3 to 20.5% in degassed methanol) and an excitation spectrum extending down to 600 nm, which made these emitters particularly promising for application in in vivo biological experiments. The experimental results and the theoretical DFT and TD DFT calculations showed that the photophysical characteristics of these complexes are largely determined by the properties of the metalating **N^C** ligands, which provides a major contribution to the formation of the lowest S_1_ and T_1_ excited states responsible for low-energy absorption and emission, respectively. All complexes showed strong lifetime dependence on the oxygen concentration, and two of them (**1** and **7**) were tested as oxygen sensors in cell cultures. We established a procedure for the hydrophobic compounds’ solubilization in aqueous solutions and obtained Stern-Volmer calibrations (lifetime vs. oxygen concentration) in the model physiological media ((DMEM/FBS/PEG-200). It was shown that Stern-Volmer quenching constants in methanol and media containing a large amount of serum albumin differ substantially, as determined by the complexes’ absorption in albumin hydrophobic pockets to give noncovalent {albumin/complexes} conjugates. The conjugates formation changed the chromophore availability for the interaction with the quencher, but still maintained the ability of the emitters’ lifetimes to respond to variations in oxygen concentration in the system under study. Colocalization and toxicity studies using the CHO-K1 cell line showed that the complexes displayed preferential localization either in mitochondria (**1**) or in lysosomes (**7**), and were not toxic at the concentration of 5.0 and 10.0 µM, respectively. Both complexes were used in the PLIM experiments, which indicated their applicability for semi-quantitative measurements of oxygen concentration in cell samples and also showed that they are promising for use in in vivo studies. 

## 4. Materials and Methods

**General comments.** The ^1^H and ^1^H-^1^H COSY and NOESY (400 MHz) NMR spectra were recorded on a Bruker 400 MHz Avance; chemical shift values were referenced to the solvent residual signals. Mass spectra were recorded on a Bruker maXis HRMS-ESI-QTOF in the ESI^+^ mode. Microanalyses was carried out at the analytical laboratory of the University of Eastern Finland.

Ligands **N^C1** [35], **N^C2** [26], and **N^N1 [28]**, ligand precursors *tert*-butyl 6-chlorophenanthridine-8-carboxylate [47], 6-chlorophenanthridine-2-carbonitrile [26], 5,6-dimethoxybenzothiophene [48], 2-(5,6-dimethoxy)benzothienylboronic acid [49], and 1,10-phenanthroline-5,6-dione [50], as well as dimeric complexes **Ir_2_(N^C1)_4_Cl_2_ [35]** and **Ir_2_(N^C2)_4_Cl_2_ [26]**, were synthesized according to the published procedures. Methanol for photophysical measurements was distilled and dried before use. Other reagents and solvents were used as received without further purification.

**Synthesis of *tert*-butyl 6-(benzothiophen-2-yl)phenanthridine-8-carboxylate.** A mixture of *tert*-butyl 6-chlorophenanthridine-8-carboxylate (277 mg, 0.882 mmol), 2-benzothienylboronic acid (180 mg, 1.01 mmol), Pd(PPh_3_)_4_ (50 mg, 0.043 mmol), 10% aq. Na_2_CO_3_ (5 mL), toluene (5 mL), and dioxane (2.5 mL) was stirred in a sealed tube under an argon atmosphere at 100 °C overnight. The mixture was cooled down to RT to give a slurry. The precipitate was filtered off, washed with little amounts of water and ethanol, and dried. The target compound was purified using column chromatography with dichloromethane/ethyl acetate 10:1 as eluent. Yield 81%, 295 mg, pale yellow solid. ^1^H NMR (CDCl_3_, 400 MHz, δ): 9.44 (s, 1H), 8.77 (d, J = 8.6 Hz, 1H), 8.65 (d, J = 8.0 Hz, 1H), 8.48 (d, J = 8.2 Hz, 1H), 8.33 (d, J = 8.2 Hz, 1H), 8.01 (s, 1H), 8.00–7.94 (m, 2H), 7.86 (t, J = 7.4 Hz, 1H), 7.76 (t, J = 7.6 Hz, 1H), 7.46 (m, 2H), 1.68 (s, 9H, tBu). HRMS (ESI) m/z: 412.1317 calcd. for C_26_H_22_NO_2_S^+^ [M + H]^+^, found 412.1308.

**Synthesis of 6-(benzothiophen-2-yl)phenanthridine-8-carboxylic acid (N^C3).** In a 25 mL flask, a solution of tert-butyl 6-(benzothiophen-2-yl)phenanthridine-8-carboxylate (412 mg, 1.00 mmol) in 10 mL of 1:1 CH_2_Cl_2_/CF_3_COOH mixture was stirred at RT for 5 h. After being evaporated, the resulting solid was diluted with water (20 mL) and neutralized carefully with 1 N NaOH until pH ≈ 6. The precipitated solid was centrifuged, washed successively with water, an ethanol/diethyl ether 1:2 mixture, and a pure diethyl ether, and dried at 100 °C to establish the product (300 mg, 84%) as a beige solid. ^1^H NMR ((CD_3_)_2_SO, 400 MHz, δ): 13.45 (br s, 1H, COOH), 9.22 (s, 1H), 9.06 (d, J = 8.6 Hz, 1H), 8.88 (d, J = 8.0 Hz, 1H), 8.46 (d, J = 8.2 Hz, 1H), 8.16 (d, J = 8.2 Hz, 1H), 8.13–8.05 (m, 3H), 7.90 (t, J = 7.4 Hz, 1H), 7.82 (t, J = 7.6 Hz, 1H), 7.51 (m, 2H). HRMS (ESI) m/z: 356.0745 calcd. for C_22_H_14_NO_2_S^+^ [M + H]^+^, found 356.0722.

**Synthesis of 6-(5,6-dimethoxybenzothiophen-2-yl)phenanthridine-2-carbonitrile.** A mixture of 6-chlorophenanthridine-2-carbonitrile (210 mg, 0.882 mmol), 2-(5,6-dimethoxy) benzothienylboronic acid (180 mg, 1.01 mmol), Pd(PPh_3_)_4_ (50 mg, 0.043 mmol), 10% aq. Na_2_CO_3_ (5 mL), toluene (5 mL), and ethanol (2.5 mL) was stirred in a sealed tube under an argon atmosphere at 100 °C for overnight. The mixture was cooled down to RT to provide a precipitate, which was filtered off, washed with water, ethanol, and diethyl ether and thoroughly vacuum-dried. Yield 78%, 272 mg, yellow solid. ^1^H NMR ((CD_3_)_2_SO, 400 MHz, δ): 9.39 (s, 1H), 9.04 (d, J = 8.3 Hz, 1H), 8.78 (d, J = 8.3 Hz, 1H), 8.02–8.20 (m, 4H), 7.93 (t, J = 7.6 Hz, 1H), 7.62 (s, 1H), 7.52 (s, 1H), 3.90 (s, 3H, OMe), 3.87 (s, 3H, OMe). HRMS (ESI) m/z: calcd for C_24_H_16_N_2_NaO_2_S^+^ [M + Na]^+^ 419.0825, found 419.0824.

**Synthesis of 6-(5,6-dimethoxybenzothiophen-2-yl)phenanthridine-2-carboxylic acid (N^C4).** A mixture of 6-(5,6-dimethoxybenzothiophen-2-yl)phenanthridine-2-carbonitrile (380 mg, 1.13 mmol), NaOH (1.0 g, 25 mmol) and ethylene glycol (20 mL) was stirred at 140 °C overnight. After being cooled and centrifuged, the resulting clear orange solution was diluted with water (40 mL) and acidified carefully with 6 N HCl until pH ≈ 6. The thick suspension formed was centrifuged, the precipitate was washed successively with water, ethanol, and ether, and dried at 100 °C. Yield 82%, 330 mg, pale brown solid. ^1^H NMR ((CD_3_)_2_SO, 400 MHz, δ): 13.29 (br s, 1H, COOH), 9.31 (s, 1H), 8.98 (d, J = 8.3 Hz, 1H), 8.78 (d, J = 8.3 Hz, 1H), 8.32–8.25 (m, 1H), 8.14 (d, J = 8.5 Hz, 1H), 8.11 (s, 1H), 8.06 (t, J = 7.5 Hz, 1H), 7.91 (t, J = 7.6 Hz, 1H), 7.63 (s, 1H), 7.53 (s, 1H), 3.90 (s, 3H, OMe), 3.87 (s, 3H, OMe). HRMS (ESI) m/z: 416.0951 calcd. for C_24_H_18_NO_4_S^+^ [M + H]^+^, found 416.0937.

**Synthesis of 4-(2-(3-bromophenyl)-1H-imidazo[4,5-f][1,10]phenanthrolin-1-yl)benzoic acid (N^N2).** 1,10-phenanthroline-5,6-dione (300 mg, 1.427 mmol), 4-aminobenzoic acid (235 mg, 1.713 mmol), 3-bromobenzaldehyde (264 mg, 1.427 mmol), ammonium acetate (132 mg, 1.713 mmol), and 10 mL of glacial acetic acid were stirred at 70 °C in a 25 mL round-bottom flask for 12 h. The reaction mixture was poured in ice and diluted with 1 N NaOH to pH ≈ 6. As a result, the beige precipitate was formed. It was washed and subsequently centrifuged with water, methanol, acetone/DMSO 4:1 mixture, pure acetone, and diethyl ether, and then vacuum-dried. Yield: 495 mg, 70%. ^1^H NMR ((CD_3_)_2_SO, 400 MHz, δ): 9.11 (d, J = 4.8 Hz, 1H), 9.05 (d, J = 8.1 Hz, 1H), 8.98 (d, J = 4.4 Hz, 1H), 8.15 (d, J = 8.1 Hz, 2H), 7.89 (m, 1H), 7.86 (m, 1H), 7.70 (d, J = 8.1 Hz, 2H), 7.62 (d, J = 8.4 Hz, 1H), 7.56–7.51 (m, 2H), 7.42 (d, J = 8.4 Hz, 1H), 7.34 (t, J = 7.8 Hz, 1H). HRMS (ESI) m/z: 517.0276 calcd. for C_26_H_15_BrN_4_NaO_2_^+^ [M + Na]^+^, found 517.0239.

**Synthesis of iridium dimeric complex Ir_2_(N^C3)_4_Cl_2_.** In a 25 mL round-bottom flask were placed IrCl_3_·6H_2_O (90 mg, 0.221 mmol), **N^C3** ligand (163 mg, 0.460 mmol), 2-ethoxyethanol (9 mL), and distilled water (3 mL). The reaction mixture was stirred under reflux for 12 h and then evaporated. The solid residue was dispersed in water and centrifuged, then dissolved in acetone and centrifuged once again. The resulting solution was evaporated; the dried residue was thoroughly washed with diethyl ether and vacuum-dried. Dark red solid, 188 mg, yield 91%. ^1^H NMR ((CD_3_)_2_SO, 400 MHz, δ): 13.67 (br s, 1H, COOH), 9.86 (s, 1H), 9.16 (d, J = 7.8 Hz, 1H), 8.81 (d, J = 8.9 Hz, 1H), 8.58 (d, J = 8.7 Hz, 1H), 8.38 (d, J = 8.9 Hz, 1H), 7.94 (d, J = 8.9 Hz, 1H), 7.61 (dd, J = 8.6, 8.2 Hz, 1H), 7.44 (dd, J = 8.2, 7.8 Hz, 1H), 7,10 (dd, J = 8.6, 8.2 Hz, 1H), 6.61 (dd, J = 8.7, 8.3 Hz, 1H), 6.43 (d, J = 8.5 Hz, 1H).

**Synthesis of iridium dimeric complex Ir_2_(N^C4)_4_Cl_2_.** In a 25 mL round-bottom flask were placed IrCl_3_·6H_2_O (90 mg, 0.221 mmol), **N^C4** ligand (191 mg, 0.460 mmol), 2-ethoxyethanol (9 mL), and distilled water (3 mL). The reaction mixture was stirred under reflux for 36 h. The resulting suspension was centrifuged to provide the precipitate, which was washed with water (1 × 5 mL), methanol (2 × 5 mL), and diethyl ether (1 × 5 mL), and then vacuum-dried. Dark brown solid, 221 mg, yield 95%. ^1^H NMR ((CD_3_)_2_SO, 400 MHz, δ): 13.15 (br s, 1H, COOH), 9.23 (s, 1H), 9.08 (d, J = 7.8 Hz, 1H), 9.04 (d, J = 8.5 Hz, 1H), 8.15 (dd, J = 8.1 Hz, 7.5 Hz, 1H), 8.09 (dd, J = 8.1 Hz, 7.5 Hz, 1H), 7.90 (d, J = 9.1 Hz, 1H), 7.40 (s, 1H), 5.81 (s, 1H), 3.69 (s, 3H, OMe), 2.34 (s, 3H, OMe).

**General procedure for the synthesis of iridium [Ir(N^C#)_2_(N^N#)][PF_6_] (1–7) complexes.** The following components were placed in a 5 mL vial: the corresponding iridium dimeric complex (0.018 mmol), **N^N#** ligand (0.038 mmol), KPF_6_ (0.380 mmol), acetone (2 mL), and DMF (3 drops, ca. 50 mg). The reaction mixture was stirred for 2 days at 40 °C in the absence of light and thoroughly evaporated. The solid residue was suspended in water and centrifuged. The solution was decanted and washed with water again, and the solid residue was dissolved several times in acetone and centrifuged. The obtained acetone solutions were combined and evaporated to dryness. The resulting solid material was thoroughly washed with diethyl ether and dried to provide the desired product.

**[Ir(N^C1)_2_(N^N1)][PF_6_] (1).** Dark red solid, 46 mg, yield 94%. ^1^H NMR ((CD_3_)_2_CO, 400 MHz, δ): 9.81 (d, J = 9.7 Hz, 1H), 9.69 (d, J = 9.5 Hz, 1H), 9.14 (d, J = 9.5 Hz, 1H), 8.96 (d, J = 5.3 Hz, 1H), 8.88 (d, J = 8.0 Hz, 1H), 8.76 (d, J = 8.8 Hz, 1H), 8.73 (d, J = 8.6 Hz, 1H), 8.64 (d, J = 8.3 Hz, 1H), 8.32 (d, J = 8.4 Hz, 1H), 8.29–8.26 (m, 2H), 8.04 (d, J = 7.7 Hz, 1H), 7.95 (d, J = 7.9 Hz, 1H), 7.86–7.80 (m, 2H), 7.72–7.58 (m, 7H), 7.38 (dd, J = 8.0, 7.0 Hz, 1H), 7.35–7.13 (m, 7H), 7.12 (d, J = 9.0 Hz, 1H), 6.96 (dd, J = 7.6, 8.6 Hz, 1H), 6.82 (d, J = 8.5 Hz, 1H), 6.78–6.72 (m, 3H), 6.62 (d, J = 8.2 Hz, 1H), 6.57 (dd, J = 8.2, 7.0 Hz, 1H), 6.39 (d, J = 8.2, 7.0 Hz, 1H), 5.49 (d, J = 8.9 Hz, 1H), 3.97 (s, 3H, OMe). HRMS (ESI) m/z: 1214.2541 calcd. for C_69_H_43_IrN_5_OS_2_^+^ [M]^+^, found 1214.2645. Anal. Calculated for C_69_H_43_F_6_IrN_5_OPS_2_: C, 60.96; H, 3.19; N, 5.15; experimental: C, 60.67; H, 3.23; N, 4.99.

**[Ir(N^C1)_2_(N^N2)][PF_6_] (2).** Dark red solid, 47 mg, yield 89%. ^1^H NMR ((CD_3_)_2_CO, 400 MHz, δ): 9.55 (d, J = 8.5 Hz, 1H), 9.51 (d, J = 8.0 Hz, 1H), 9.35 (d, J = 5.2 Hz, 1H), 9.21 (d, J = 5.2 Hz, 1H), 9.17 (d, J = 8.3 Hz, 1H), 8.81 (dd, J = 9.0, 8.6 Hz, 2H), 8.48 (d, J = 8.3 Hz, 1H), 8.43 (d, J = 8.3 Hz, 1H), 8.27 (m, 2H), 8.19 (d, J = 8.4, 7.6 Hz, 1H), 8.18–8.09 (m, 4H), 8.08 (d, J = 8.1 Hz, 1H), 8.05 (d, J = 8.3 Hz, 1H), 7.81 (dd, J = 7.6, 6.0 Hz, 1H), 7.77 (d, J = 8.1 Hz, 1H), 7.73 (d, J = 8.2 Hz, 1H), 7.65 (d, J = 8.2 Hz, 2H), 7.48 (d, J = 8.8 Hz, 1H), 7.38 (d, J = 8.7 Hz, 2H), 7.31 (dd, J = 7.8, 7.4 Hz, 1H), 7.27–7.21 (m, 3H), 6.91 (dd, J = 8.7, 8.3 Hz, 2H), 6.86 (dd, J = 8.2, 8.0 Hz, 1H), 6.77 (d, J = 8.8 Hz, 2H), 6.74–6.69 (m, 3H). HRMS (ESI) m/z: 1307.1376 calcd. for C_68_H_39_BrIrN_6_O_2_S_2_^+^ [M]^+^, found 1307.1417. Anal. Calculated for C_68_H_39_BrF_6_IrN_6_O_2_PS_2_: C, 56.20; H, 2.71; N, 5.78; experimental: C, 55.98; H, 2.61; N, 5.47.

**[Ir(N^C1)_2_(N^N3)][PF_6_] (3).** Dark red solid, 37 mg, yield 90%. ^1^H NMR ((CD_3_)_2_CO, 400 MHz, δ): 9.51 (d, J = 8.3 Hz, 1H), 9.24 (m, 1H), 8.89 (dd, J = 7.0, 6.2 Hz, 2H), 8.71 (m, 1H), 8.53 (d, J = 8.2 Hz, 1H), 8.47 (d, J = 8.2 Hz, 1H), 8.39 (d, J = 7.9 Hz, 1H), 8.17 (dd, J = 8.6, 8.0 Hz, 1H), 8.14 (dd, J = 8.6, 8.0 Hz, 1H), 8.07 (d, J = 8.1 Hz, 1H), 8.05–7.95 (m, 4H), 7.92 (d, J = 8.8 Hz, 1H), 7.72 (dd, J = 7.2, 6.8 Hz, 1H), 7.56 (m, 1H), 7.37 (d, J = 8.7 Hz, 1H), 7.36–7.21 (m, 6H), 7.08 (m, 1H), 6.98 (d, J = 8.2 Hz, 1H), 6.87 (d, J = 8.2 Hz, 1H), 6.83 (d, J = 8.4, 8.0 Hz, 1H), 6.72 (dd, J = 7.8, 7.4 Hz, 2H), 6.41 (dd, J = 8.2, 7.8 Hz, 1H). HRMS (ESI) m/z: 1008.1808 calcd. for C_54_H_33_IrN_5_S_2_^+^ [M]^+^, found 1008.1806. Anal. Calculated for C_54_H_33_F_6_IrN_5_PS_2_: C, 56.24; H, 2.88; N, 6.07; experimental: C, 56.01; H, 2.83; N, 5.84.

**[Ir(N^C1)_2_(N^N4)][PF_6_] (4).** Dark red solid, 38 mg, yield 92%. ^1^H NMR ((CD_3_)_2_CO, 400 MHz, δ): 9.53 (d, J = 8.1 Hz, 2H), 9.31 (d, J = 5.3 Hz, 2H), 8.79 (d, J = 8.0 Hz, 2H), 8.64 (d, J = 8.3 Hz, 2H), 8.41 (d, J = 8.2 Hz, 2H), 8.18 (dd, J = 7.4, 7.0 Hz, 2H), 8.12 (dd, J = 8.2, 7.6 Hz, 2H), 8.09 (d, J = 8.3 Hz, 2H), 8.07 (d, J = 8.1 Hz, 2H), 7.98 (s, 2H), 7.52 (d, J = 8.6 Hz, 2H), 7.25 (dd, J = 7.4, 7.0 Hz, 2H), 7.23 (dd, J = 7.8, 7.4 Hz, 2H), 6.96 (d, J = 8.2 Hz, 2H), 6.75 (dd, J = 9.2, 8.9 Hz, 2H), 6.73 (dd, J = 8.4. 8.0 Hz, 2H). HRMS (ESI) m/z: 993.1699 calcd. for C_54_H_32_IrN_4_S_2_^+^ [M]^+^, found 993.1704. Anal. Calculated for C_54_H_32_F_6_IrN_4_PS_2_: C, 56.99; H, 2.83; N, 4.92; experimental: C, 56.22; H, 2.92; N, 4.64.

**[Ir(N^C2)_2_(N^N2)][PF_6_] (5).** Dark red solid, 47 mg, yield 91%. ^1^H NMR ((CD_3_)_2_CO, 400 MHz, δ): 9.88 (d, J = 9.1 Hz, 1H), 9.71 (d, J = 7.0 Hz, 1H), 9.41 (s, 1H), 9.24 (d, J = 6.7 Hz, 1H), 9.19 (s, 1H), 8.99 (d, J = 5.8 Hz, 1H), 8.78 (d, J = 8.5 Hz, 1H), 8.75 (d, J = 8.5 Hz, 1H), 8.45 (d, J = 7.7 Hz, 1H), 8.35 (m, 2H), 7.99 (d, J = 4.2 Hz, 1H), 7.97 (d, J = 4.2 Hz, 1H), 7.87 (d, J = 8.0 Hz, 1H), 7.85 (dd, J = 8.0, 7.6 Hz, 1H), 7.72 (d, J = 8.4 Hz, 1H), 7.70–7.63 (m, 4H), 7.61 (dd, J = 7.8, 7.4 Hz, 1H), 7.50 (d, J = 9.0 Hz, 1H), 7.46 (dd, J = 8.2, 7.8 Hz, 1H), 7.39 (dd, J = 7.8, 7.4 Hz, 1H), 7.31 (d, J = 8.7 Hz, 1H), 7.29–7.24 (m, 3H), 7.19 (dd, J = 7.8, 7.4 Hz, 1H), 6.92 (d, J = 8.5 Hz, 1H), 6.89 (d, J = 8.2 Hz, 1H), 6.80 (dd, J = 7.8, 7.4 Hz, 1H), 6.77 (d, J = 8.4 Hz, 1H), 6.75 (d, J = 8.0 Hz, 1H), 6.68 (d, J = 8.2 Hz, 1H), 6.62 (dd, J = 8.0, 7.6 Hz, 1H), 6.41 (dd, J = 7.8, 7.4 Hz, 1H), 5.45 (d, J = 8.9 Hz, 1H), 3.96 (s, 3H, OMe). HRMS (ESI) m/z: 1302.2338 calcd. for C_71_H_43_IrN_5_O_5_S_2_^+^ [M]^+^, found 1302.2437. Anal. Calculated for C_71_H_43_F_6_IrN_5_O_5_PS_2_: C, 58.92; H, 2.99; N, 4.84; experimental: C, 58.38; H, 3.07; N, 4.82.

**[Ir(N^C3)_2_(N^N1)][PF_6_] (6).** Dark red solid, 46 mg, yield 88%. ^1^H NMR ((CD_3_)_2_CO, 400 MHz, δ): 10.28 (s, 1H), 9.68 (d, J = 8.8 Hz, 1H), 9.25 (d, J = 9.4 Hz, 1H), 8.94 (d, J = 8.2 Hz, 1H), 8.73–8.66 (m, 3H), 8.61 (d, J = 8.9 Hz, 1H), 8.51 (d, J = 8.6 Hz, 1H), 8.44 (d, J = 8.9 Hz, 1H), 8.23 (s, 1H), 8.07 (d, J = 8.0 Hz, 1H), 7.94 (d, J = 8.2 Hz, 1H), 7.86 (dd, J = 8.4, 8.0 Hz, 1H), 7.79–7.75 (m, 2H), 7.71–7.61 (m, 4H), 7.58–7.51 (m, 2H), 7.31–7.13 (m, 5H), 7.06 (d, J = 8.7 Hz, 1H), 7.02 (dd, J = 7.6, 7.2 Hz, 1H), 6.95 (d, J = 8.6 Hz, 1H), 6.78 (dd, J = 8.2, 7.8 Hz, 1H), 6.58 (m, 2H), 6.52 (d, J = 8.0 Hz, 2H), 6.40 (d, J = 8.6 Hz, 1H), 6.25 (dd, J = 7.6, 7.2 Hz, 1H), 5.22 (d, J = 8.7 Hz, 1H), 3.92 (s, 3H, OMe). HRMS (ESI) m/z: 1302.2338 calcd. for C_71_H_43_IrN_5_O_5_S_2_^+^ [M]^+^, found 1302.2340. Anal. Calculated for C_71_H_43_F_6_IrN_5_O_5_PS_2_: C, 58.92; H, 2.99; N, 4.84; experimental: C, 58.33; H, 3.17; N, 4.69.

**[Ir(N^C4)_2_(N^N2)][PF_6_] (7).** Dark red-brown solid, 52 mg, yield 93%. ^1^H NMR ((CD_3_)_2_CO, 400 MHz, δ): 9.94 (d, J = 9.2 Hz, 1H), 9.60 (d, J = 7.5 Hz, 1H), 9.38 (s, 1H), 9.18 (d, J = 7.6 Hz, 1H), 9.14 (s, 1H), 9.04 (d, J = 6.1 Hz, 1H), 8.79 (d, J = 8.6 Hz, 1H), 8.75 (d, J = 8.4 Hz, 1H), 8.39 (d, J = 8.4 Hz, 1H), 8.29 (m, 2H), 8.08 (d, J = 8.4 Hz, 1H), 7.89 (dd, J = 8.0, 7.6 Hz, 1H), 7.72 (d, J = 9.3 Hz, 1H), 7.70–7.61 (m, 4H), 7.59 (d, J = 7.9 Hz, 1H), 7.54 (d, J = 8.7 Hz, 1H), 7.49 (s, 1H), 7.43 (dd, J = 7.5, 7.1 Hz, 1H), 7.37 (s, 1H), 7.37–7.31 (m, 2H), 7.25 (dd, J = 7.8, 7.4 Hz, 1H), 7.24 (d, J = 8.5 Hz, 1H), 6.90 (d, J = 9.1 Hz, 1H), 6.84 (d, J = 8.0 Hz, 1H), 6.76 (d, J = 7.9 Hz, 1H), 6.64 (dd, J = 7.6, 7.2 Hz, 1H), 6.24 (s, 1H), 6.11 (s, 1H), 5.37 (d, J = 9.2 Hz, 1H), 3.96 (s, 3H, OMe), 3.84 (s, 3H, OMe), 3.80 (s, 3H, OMe), 2.67 (s, 3H, OMe), 2.25 (s, 3H, OMe). HRMS (ESI) m/z: 1422.2761 calcd. for C_56_H_36_IrN_6_O_8_S_2_^+^ [M]^+^, found 1422.2830. Anal. Calculated for C_75_H_51_F_6_IrN_5_O_9_PS_2_: C, 57.47; H, 3.28; N, 4.47; experimental: C, 57.23; H, 3.36; N, 4.25.

**Photophysical experiments.** Photophysical measurements in solution were performed in aqueous media and, partially, in methanol. Absorption spectra were measured with a Shimadzu UV-1800 spectrophotometer. The excitation spectra in solution were recorded using a Fluorolog-3 (HORIBA Jobin Yvon) spectrofluorimeter. The emission spectra were registered using an Avanted AvaSpec-2048x64 spectrometer. The absolute emission quantum yield was determined in solution by a comparative method. LED (365 nm) was used for pumping and [Ru(bpy)_3_][PF_6_]_2_ water solution (Φ = 0.040 air-saturated, 0.063 Ar-saturated) was used as the reference. A pulse laser DTL-355 Basic (wavelength 355 nm, pulse width 5 ns, repetition frequency 1000 Hz), a Hamamatsu (H10682-01) photon counting head, FASTComTec (MCS6A1T4) multiple-event time digitizer, and an Ocean Optics monochromator (Monoscan-2000, interval of wavelengths 1 nm) were used for lifetime measurements. An oxygen meter (PyroScience FireStingO2, equipped with an oxygen probe OXROB10 and a temperature sensor TDIP15) was used to determine partial pressure and concentration of molecular oxygen in aqueous solutions. Temperature control was performed by using a Quantum Northwest qpod-2e cuvette sample compartment.

**Computational Details.** All calculations were performed using the Gaussian 16 [51] computer code in DFT methodology. A hybrid Austin-Frisch-Petersson functional with dispersion (APFD) [52] was used with the Pople’s Gaussian-type function basis sets [53] 6-311+G* on heteroatoms, 6-31G* for carbon and hydrogen atoms. The Stuttgart-Dresden effective core pseudopotential and the corresponding basis set were used for iridium [54]. The Polarizable Continuum Model (PCM) was applied to account for non-specific solvation [55]. Full geometry optimization was performed for all compounds under consideration (**1** and **7**). Emission energies were obtained as differences between the energies of the optimized triplet and the singlet states. The electronic absorption spectra were calculated within TD-DFT with 200 excited states. The convoluting of UV/Vis spectra from calculated oscillator strengths were obtained using the method described in reference [56] and modified for Lorentzian broadening. A qualitative picture of the displacement of the electron density during absorption and emission transitions was established by the construction of Natural Transition Orbitals (NTO) [57]. The changes in electronic density *Δρ* during the *S*_0_ →*S_i_* transitions were calculated as:$$\Delta \rho \left(S_{0}\to S_{i}\right)=\underset{k}{\sum }{\left|\Psi_{ik}\left(virt\right)\right|}^{2}-\underset{k}{\sum }{\left|\Psi_{ik}\left(occ\right)\right|}^{2}$$
where *Ψ_ik_*(*occ*) and *Ψ_ik_*(*virt*) are NTO pairs for *S*_0_ → *S_i_* transition. The electronic density’s change during *T_1_*→*S*_0_ transition was calculated in analogous manner using canonical Kohn–Sham HSOMO-α (highest single occupied molecular α-spin orbital) and LSUMO-β (lowest single unoccupied molecular β-spin orbital). A quantitative estimation of electrons transferred between the parts of the molecules was obtained by the IFCT (Interfragment charge transfer) method [58]. The Multiwfn 3.6 program [58] was used for both methods.

**Preparation of aqueous solutions of complexes 1 and 7.** To the solution of complex **1** or **7** in PEG-200 (22.5 or 37.5 µL, concentration of complexes are 333 µM and 400 µM, respectively) the appropriate quantity of (1) PBS or (2) 50 µM solution of BSA in PBS, or (3) DMEM, containing 10% of FBS, was added (1477.5 or 1462.5 µL, respectively), followed by vigorous stirring for 1 min and subsequent incubation at 37 °C for 24 h. Thus, for complex **1** PBS-PEG(200), PBS-BSA-PEG(200) and DMEM-FBS-PEG(200) systems with the concentration of **1** = 5 µM and volume content of PEG-200 = 1.5% were obtained. For complex **7**, the PBS-PEG(200), PBS-BSA-PEG(200), and DMEM-FBS-PEG(200) systems had a concentration of **7** = 10 µM and a volume content of PEG-200 = 2.5%.

The PBS concentration used for photophysical measurements was 0.01 M, C_NaCl_ = 0.14 M, pH = 7.4. The PBS concentration used for analytical chromatography (GPC) was 0.01 M, C_NaCl_ = 0.14 M, pH = 6.8.

**Analytical chromatography of the mixtures of complexes 1 and 7 with BSA.** The samples of the above-described PBS-BSA-PEG(200) solutions of complexes **1** and **7** were used for these measurements. Analytical gel-permeation chromatography (GPC) was performed on a “Prominence 20” chromatograph (“Shimadzu”, Japan) equipped with a «PSS PROTEEMA analytical 100 Å» column (300 × 8 mm, 5 µm) and UV-vis (SPD-M20A) detector. Chromatography was carried out at 30 °C using 0.01 M PBS (pH 6.8; C_NaCl_ = 0.2 M) as an eluent at 0.5 mL/min flow rate. Sample aliquots were 5 µL.

**Cell Culturing.** The Chinese hamster ovary CHO-K1 cells were maintained in DMEM/F12 (Biolot, St. Petersburg, Russia) medium supplemented with 10% FBS (Gibco, Carlsbad, CA, USA), 2 mM glutamine (Gibco, Carlsbad, CA, USA), and penicillin/streptomycin at a concentration 100 U/mL (Thermo Fisher Scientific, Waltham, MA, USA). The cells were maintained in a humidified incubator at 37 °C with 5% CO_2_ and passaged routinely using trypsin-EDTA (Thermo Fisher Scientific, Waltham, MA, USA). For live-cell confocal microscopy, the cells in concentrations of 0.5 × 10^5^ CHO-K1 cells per 1 mL of growing media were seeded in glass bottom 35 mm dishes (Ibidi GmbH, Gräfelfing, Germany) and incubated for 48 h until reaching a confluence of 60–70%. Complexes **1** and **7** were dissolved in PEG-200 at a concentration of 1 mM, mixed with a required volume of PEG-200, and diluted with the growing media to the concentration of 5 µM for complex **7** and 10 µM for complex **1**. After incubation with the probe for 24 h, the cells were rinsed with fresh media with all supplements.

**Colocalization study.** For the vital staining of mitochondria in CHO-cells, the cells were incubated with BioTracker 405 Blue Mitochondria Dye (Sigma-Aldrich, Merck, Munich, Germany) at the concentration of 50 nM for 15 min. For the vital staining of lysosomes and late endosomes, LysoTracker Green DND-26 (Thermo Fisher Scientific, Waltham, MA, USA) was used at the concentration of 50 nM, incubation time 30 min. Hoechst 33342 (Thermo Fisher Scientific, Waltham, MA, USA) at a concentration of 2 nM was used for vital staining of the nucleus.

**MTT Assay.** Cytotoxicity of the complexes was estimated by using the MTT protocol. CHO-K1 cells were seeded in 96-well plates (Nunc, Thermo Fisher Scientific, Waltham, MA, USA), 1 × 10^4^ cells in 100 µL of culture medium/well, and incubated overnight. The complexes were dissolved in DMSO or PEG200 at a concentration of 1 mM, mixed with supplemented media, and added to the cells to a final concentration of 0–150 µM. After incubation for 24 h, the cells were treated with MTT reagent 3(4,5-dimethyl-2-thiasolyl)-2,5-diphenyl-2H-tetrasole bromide (Thermo Fisher Scientific, Waltham, MA, USA) at the concentration of 0.5 mg/mL according to the manufacturer’s protocol. After further incubation at 37 °C under 5% CO_2_ for 3 h, the media were removed, and the formazan crystals were dissolved in 100 µL of DMSO (Merck, Munich, Germany). The absorbance in each well was measured at 570 nm using a SPECTROstar Nano microplate reader (BMG LABTECH, Ortenberg, Germany). Viability was determined as a ratio of the average absorbance value of the wells containing conjugate to that of the control. The results were shown as mean ± standard deviation from 5–12 repetitions.

**Confocal Microscopy.** The imaging of live CHO-K1 cells was carried out by using a confocal inverted Nikon Eclipse Ti2 microscope (Nikon Corporation, Tokyo, Japan) with x60 oil immersion objective. The required temperature and 5% CO_2_ during the experiments were maintained by using a Stage Top Incubator Tokai HIT (Japan) equipped with digital gas mixer GM-8000. The emission of the probes was excited at 405 nm, and the emission of the complexes was registered in red (570–620 nm) channels. Fluorescence of Hoechst 33342 and BioTracker 405 Blue was excited at 405 nm and recorded at 425–475 nm. LysoTracker Green was excited at 488 nm and recorded at 500–550 nm. Differential interference contrast (DIC) images were recorded in addition to the fluorescence microphotographs. The images were processed and analyzed using ImageJ software (National Institutes of Health, Bethesda, MY, USA). ImageJ JACoP Plugin was used for the quantitative co-localization analysis and the determination of Pearson (P) and Manders’ (M1 and M2) co-localization coefficients. Thresholds for M1 and M2 calculations were set by a visually estimated value for each channel. The results were presented as mean ± standard deviation of ca. 50 cells.

**Phosphorescence lifetime imaging.** Phosphorescence lifetime imaging microscopy (PLIM) of CHO-K1 cells was performed using the Nikon Eclipse Ti2 confocal devise equipped with the time-correlated single photon counting (TCSPC) DCS-120 module (Becker&Hickl GmbH, Berlin, Germany). The emission of the probes was excited with picosecond laser (405 nm) and the phosphorescence was detected in the range 690–750 nm using the 720/60 nm band pass filter. Oil immersion 60× objective with zoom 5.33 provided a scan area of 0.05 mm × 0.05 mm. The following image acquisition settings were used for complex **1**: frame time 14.53 s, pixel dwell time 54.90 µs, points number 1024, time per point 50.00 ns, time range of PLIM recording 51.20 µs, and total acquisition time 110–150 s. The following image acquisition settings were used for complex **7**: frame time 21.61 s, pixel dwell time 81.90 µs, points number 1024, time per point 75.00 ns, time range of PLIM recording 76.80 µs, and total acquisition time 120–150 s. Image size was 512 × 512 pixels. Phosphorescence lifetime distribution was calculated using SPCImage 8.1 software (Becker & Hickl GmbH, Berlin, Germany). The phosphorescence decay curves were fitted in monoexponential decay mode with an average goodness of the fit 0.8 ≤ χ^2^ ≤ 1.2. The starting fragment of the decays was omitted (1 µs for complex **1**, 5 µs for complex **7**). The average number of photons per curve was not less than 2500 at binning 8–9.

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
