# Peer review of "Novel NIR-Phosphorescent Ir(III) Complexes: Synthesis, Characterization and Their Exploration as Lifetime-Based O2 Sensors in Living Cells"

_molecules, 2022, doi:10.3390/molecules27103156_

Round 1

Reviewer 1 Report

Kritchenkov reported a series of [Ir(N^C)2(N^N)]+ NIR emitting orthometalated complexes. They have structurally characterized using elemental analysis, mass-spectrometry and NMR spectroscopy. They also demonstrated their usage in lifetime-based O2 sensing. This work should be of broad interest to Molecules readers. However, the lack of detail in mechanism discussion is a key weakness of the ms. It may be finally acceptable, but must be significantly improved before its publication.

  1. A detailed discussion on the effect of the ligand structure on the quantum yield of 7 different dyes. In addition, since different ligands are used to prepare dyes, the authors are suggested to explain why the fluorescence wavelength are the same
  2. The authors are suggested to compare the quantum yield of as-prepared Ir(III) complexes with commercial Ir complexes.
  3. The authors should also evaluate the cell toxicity of 1 and 7 without PEG.
  4. The selectivity of the oxygen sensor should be demonstrated.

Author Response

Dear Editor,

We appreciate greatly the comments from both reviewers, which are evidently aimed at improvement of the manuscript. Following the reviewers’ suggestions, we made the corresponding corrections and gave point-by-point responses to all comments, which are given in red font below. To make clear the changes in the main text we are providing the manuscript with the changes highlighted with yellow marker (MS_revised_corrections_highlighted.doc).

We hope strongly that the manuscript in the present state suits completely for publication in Molecules and look forward to your decision.

Best regards,

on behalf of the authors                                                                                S. Tunik

Reviewer 1

Kritchenkov reported a series of [Ir(N^C)2(N^N)]+ NIR emitting orthometalated complexes. They have structurally characterized using elemental analysis, mass-spectrometry and NMR spectroscopy. They also demonstrated their usage in lifetime-based O2 sensing. This work should be of broad interest to Molecules readers. However, the lack of detail in mechanism discussion is a key weakness of the ms. It may be finally acceptable, but must be significantly improved before its publication.

  1. A detailed discussion on the effect of the ligand structure on the quantum yield of 7 different dyes.

In fact, similarity of the N^C ligands structure, their primary role in emission characteristics and negligible effect of substituents result in essentially similar emission energy for all seven complexes that is clearly described in the text. Closely analogous situation is observed for the quantum yields with the only exception of the complex 7, which displays substantially higher QY. To explain this observation we calculated rate constants of radiative and nonradiative excited state relaxation for all complexes (these data are now added to Table 1) and also discussed the observed trends in the text (p. 5 lines 21 – 29):

“Interestingly, emission efficiency (QY) is nearly the same for complexes 1-6 and sharply different for 7 increasing for ca. 60% compared to the other complexes. The data on radiative and nonradiative rate constants (Table 1) indicate that in the case of 7 one can observe a substantial decrease in the rate of nonradiative emissive excited state relaxation while radiative rate constants are very similar for the emitters 1, 5–7, which contain the same N^N1 ligand and differ in the structure of N^C ligands only. This observation may be explained by “rigidification” of molecular structure in 7 caused by the introduction of four –OMe substituents in the N^C4 ligands that evidently limits possible distortions in emissive triplet compared to the ground state and thus reduces so-called Huang-Rys factor,[36,37] which determines knr magnitude of phosphorescent emitters. Note that intramolecular noncovalent interactions of OMe substituents with the other components of ligand environment were also observed in the NOESY spectra of 7 (see Figure S7b).”

In addition, since different ligands are used to prepare dyes, the authors are suggested to explain why the fluorescence wavelength are the same.

The discussion concerning similarity of the phosphorescence band profiles for all studied complexes was already presented in the manuscript, see p.5, lines 12-18:

“The obtained experimental data and results of theoretical calculations led to several important conclusions: the N^N ligands do not participate in emissive excited state formation and negligibly affect the magnitude of the T1-S0 energy gap, the properties of the N^C benzothienyl-phenanthridine aromatic system have a key impact on the emissive energy gap. It is also worth noting that essentially different substituents (cf. –COOH and –OMe) inserted into the benzothienyl-phenanthridine fragment of the N^C ligands also do not perturb the emission energy evidently due to a large “capacity” of this electronic reservoir.”

We can hardly add something else to these straightforward arguments. The larger reservoir, the weaker response onto external disturbance.

  1. The authors are suggested to compare the quantum yield of as-prepared Ir(III) complexes with commercial Ir complexes.

It should be noted that comparison of QYs for different emitters only makes sense if the luminophores display essentially similar emission wavelength because emission efficiency is strongly determined by “energy gap law” and sharply decreased with the red shift of emission band into NIR region (see, for example, [10]). This is why it does not make sense to compare the QYs of the synthesized compounds with “commercial Ir complexes”. Moreover, the phosphorescence QY also depends on the media (solid state vs solution) and the nature of the solvent used for the measurements. Nevertheless, we added one sentence (p.5, lines 19-21) with the references to the other NIR emitting iridium complexes:

“The phosphorescence quantum yields are rather high for the NIR emitting phosphors (10.3-20.5% in degassed solution) and fit well the other data obtained earlier for the complexes of this type.[10,26–28,30–35]”

  1. The authors should also evaluate the cell toxicity of 1 and 7 without PEG.

Cell toxicity of a compound may be estimated only if it is soluble in the media used for the compound incubation into cell culture (typically - aqueous solution containing DMEM, FBS, buffer salts). For the water-insoluble hydrophobic compounds (as those we used in this study) it is necessary to use solubilizing agent (DMSO, PEG…) to deliver the complexes to cells. Therefore, it is impossible to perform the experiments suggested by the reviewer.

  1. The selectivity of the oxygen sensor should be demonstrated.

Indeed, this is a very important point for the whole study. Crosstalk with the other microenvironment parameters may crucially tamper the calibration of lifetime vs oxygen concentration. However, all necessary information has been already given in the manuscript. It is well-known that temperature, biomolecules and some cations (Ca2+, Mg2+) presented in biosystems may affect lifetime readings and distort the calibrations obtained under different conditions. Nevertheless, the calibration curves shown in Figure 3 were obtained at 37oC (typical temperature applied in luminescent microscopy of cell cultures). Note that to obtain these calibrations we also used the model physiological media (DMEM, FBS, buffer salts), which contains biomolecules, a wide range of metal cations typical for biosystems and effectively simulates the microenvironment in cell cultures. This is why these calibrations may be applied for semi-quantitative oxygen pressure measurements in cell cultures that has been also confirmed by PLIM experiments with CHO-K1 cell line, see Figure 7, where lifetime readings under normoxia and hypoxia fit well the values obtained in model media used for calibration.

Reviewer 2 Report

The article by Kritchenkov et al is devoted to the synthesis and study of new cyclometallated complexes of iridium(III) with ligands based on phenanthridine, as well as testing these compounds as sensors for determining molecular oxygen in living cells.

The work was done at a high level, contains numerous measurements and all the necessary research methods (possibly with the exception of X-ray diffraction analysis of single crystals) are given. The work is well written and I will separately note very clear and high-quality illustrations (for example, scheme 1).

Reviewer's notes:

  1. It is necessary to carefully check the text for typos. For example, the paragraphs "results and discussion" and "experimental section" are written with a lowercase letter. The last paragraph (“conclusion”) is more reasonable to call “conclusions”.
  2. The luminescence decay curves must be entered in the Supplementary along with the fitting curves and the approximation confidence parameters.
  3. On Figure 2, the initial intensity of the emission spectrum (about 600 nm) has a negative value, which is devoid of physical meaning.
  4. In Table 1, it is necessary to add the measurement errors of the lifetimes and quantum yields of luminescence.
  5. References to the literature in the introduction are not given in the best way. For example in a sentence «Among available phosphorescent chromophores, the complexes of transition metals such as Ru(II), Pd(II), Re(I), Ir(III), and Pt(II) are most often used for monitoring of oxygen concentration, since these coordination compounds exhibit bright luminescence in various regions of the visible and near infrared (NIR) spectrum.[10,14–17]» it is required to clarify which particular publications are devoted to complexes of ruthenium, palladium, etc.
  6. In the SI 2D 1H-1H NMR spectra are shown, but there is no assignment of the 1H spectrum signals, although, apparently, these measurements were made for this assignment. Appropriate references should be added.

Author Response

Dear Editor,

We appreciate greatly the comments from both reviewers, which are evidently aimed at improvement of the manuscript. Following the reviewers’ suggestions, we made the corresponding corrections and gave point-by-point responses to all comments, which are given in red font below. To make clear the changes in the main text we are providing the manuscript with the changes highlighted with yellow marker (MS_revised_corrections_highlighted.doc).

We hope strongly that the manuscript in the present state suits completely for publication in Molecules and look forward to your decision.

Best regards,

on behalf of the authors                                                                                S. Tunik

Reviewer 2

The article by Kritchenkov et al is devoted to the synthesis and study of new cyclometallated complexes of iridium(III) with ligands based on phenanthridine, as well as testing these compounds as sensors for determining molecular oxygen in living cells.

The work was done at a high level, contains numerous measurements and all the necessary research methods (possibly with the exception of X-ray diffraction analysis of single crystals) are given. The work is well written and I will separately note very clear and high-quality illustrations (for example, scheme 1).

Reviewer's notes:

  1. It is necessary to carefully check the text for typos. For example, the paragraphs "results and discussion" and "experimental section" are written with a lowercase letter. The last paragraph (“conclusion”) is more reasonable to call “conclusions”.

We did our best to eliminate all typos in the revised manuscript, including those mentioned by the reviewer.

  1. The luminescence decay curves must be entered in the Supplementary along with the fitting curves and the approximation confidence parameters.

The decay curves have been added to ESI, see Figures S23-S54.

  1. On Figure 2, the initial intensity of the emission spectrum (about 600 nm) has a negative value, which is devoid of physical meaning.

The Figure 2 has been corrected.

  1. In Table 1, it is necessary to add the measurement errors of the lifetimes and quantum yields of luminescence.

The corresponding values of experimental uncertainties are added to the notes for Table 1.

  1. References to the literature in the introduction are not given in the best way. For example in a sentence «Among available phosphorescent chromophores, the complexes of transition metals such as Ru(II), Pd(II), Re(I), Ir(III), and Pt(II) are most often used for monitoring of oxygen concentration, since these coordination compounds exhibit bright luminescence in various regions of the visible and near infrared (NIR) spectrum.[10,14–17]» it is required to clarify which particular publications are devoted to complexes of ruthenium, palladium, etc.

The references have been sorted out correspondingly.

  1. In the SI 2D 1H-1H NMR spectra are shown, but there is no assignment of the 1H spectrum signals, although, apparently, these measurements were made for this assignment. Appropriate references should be added.

Assignment of the signals has been done and shown in the corresponding figures in ESI (Figures S1-S7).

Round 2

Reviewer 1 Report

Recommend to publish